# Mitigation of Gravity Segregation by Foam to Enhance Sweep Efficiency

**Meijie Wang [1], Rigu Su [1,*], Yaowei Huang [2], Dengya Chen [1], Yiyang Li [1], Hong Xiang [1], Wenxuan Guo [1] and Long Chen [1]**

[1]  Xinjiang Oilfield Company Engineering and Technology Research Institute (Supervision Company), Karamay 834000, China; yc_chdy@petrochina.com.cn (D.C.); cyycl@petrochina.com.cn (L.C.)

[2]  State Key Laboratory of Oil and Gas Resources and Exploration, China University of Petroleum (Beijing), Beijing 102249, China

*  Correspondence: surigu@petrochina.com.cn

**Abstract:** Foam-assisted gas injection exhibits promising potential for enhancing sweep efficiency through the amelioration of gravity segregation, particularly within reservoirs characterized by heterogeneity. In this work, the implicit-texture (IT) model featuring two flow regimes is employed to examine the impact of heterogeneity on gravity segregation. The validation of the numerical results for water–gas coinjection and pre-generated foam injection is accomplished through a comparative analysis with analytical solutions. A hypothetical two-layer model with varying permeabilities and thickness ratios is used to examine the impact of foam on gravity segregation. The numerical findings demonstrate satisfactory conformity with analytical solutions in homogeneous reservoirs. A high-permeability top layer in a layered model with a fixed injection rate results in sweep efficiency similar to that of a homogeneous reservoir with each individual permeability. A low-permeability top layer could increase the sweep efficiency, but with severe permeability contrast, the bottom high-permeability layer could impact the displacement process, even with a thin thickness. The sweep efficiency increases with the thickness of the high-permeability top layer and decreases with a thicker low-permeability top layer under fixed injection pressure. The predicted segregation length through a single-layer approximation cannot match the results of the layered models where the permeability contrast is too great or the thickness of two layers is comparable.

**Keywords:** $CO_2$ foam; implicit-texture model; gravity segregation; sweep efficiency; layered reservoirs

## 1. Introduction

Gas injection has demonstrated its potential in improving oil recovery in low-permeability reservoirs, where water injection is challenging [1,2]. The injected gas, which includes natural gas and $CO_2$, increases pressure and displaces more oil towards the production wells. However, gas injection often faces unsatisfactory sweep efficiency due to factors such as reservoir heterogeneity, gravity override, and viscous instability. Fortunately, the utilization of foam, which consists of gas bubbles interconnected by delicate liquid films, can effectively surmount these challenges and enhance the sweep efficiency of gas injection enhanced oil recovery (EOR) techniques [3–6]. Due to this ability, foam is widely used in conventional or unconventional reservoirs to overcome the poor sweep efficiency of gas injection [7–9]. Furthermore, foam can expand the storage capacity for $CO_2$ sequestration in aquifers [10–14].

In the context of water–gas coinjection, the phenomenon of gravity segregation may manifest, whereby the injected gas ascends to the uppermost regions of the reservoir to displace oil, while the injected water descends to the lower sections and facilitates the movement of oil towards the production wells. A steady-state analytical model was presented by [15] to describe gravity segregation during simultaneous and uniform water

and gas coinjection into horizontal, homogeneous reservoirs. This model can also be employed in the context of water-alternating-gas (WAG) injection, under the assumption that the injection cycles are sufficiently short to guarantee thorough amalgamation of all slugs in close proximity to the well. As illustrated in Figure 1, the model posits the existence of three distinct regions characterized by uniform saturation within a reservoir, wherein discernible demarcations separate them: an overriding zone housing solely gas, an underride zone exclusively containing water, and an adjacent mixed zone near the wellbore where simultaneous gas and water flow coexist. This model can be utilized to estimate the maximum sweep efficiency during water–gas coinjection or WAG to enhance oil recovery. This model can also provide valuable insights for real-field applications by predicting the vertical distribution of different fluids within reservoirs, which helps optimize oil recovery strategies.

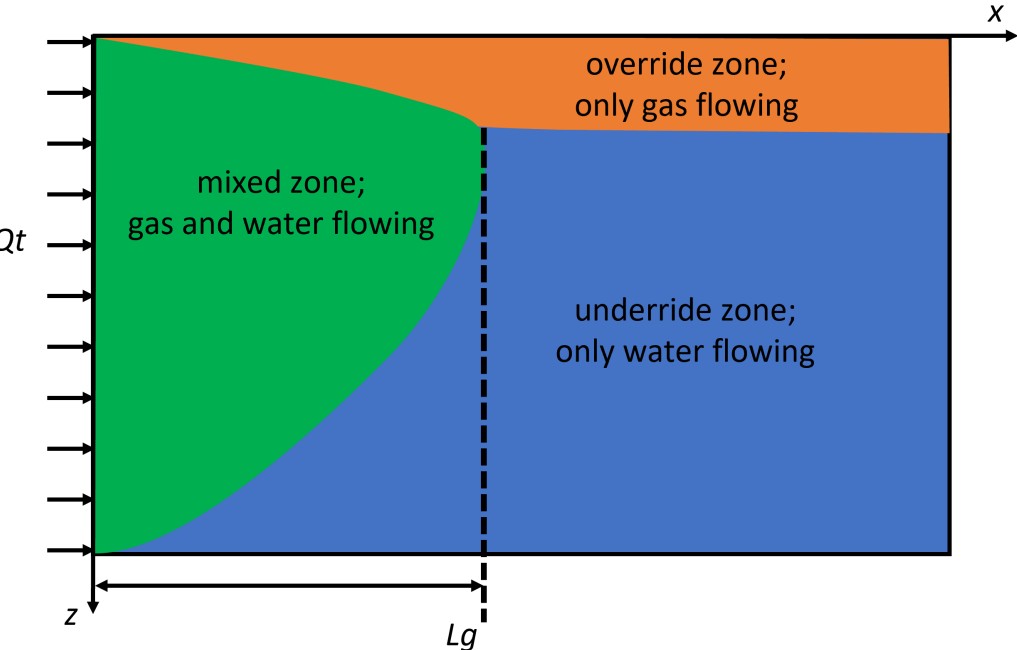

**Figure 1.** Representations of the three distinct uniform regions within the gravity-segregation model, which is employed to analyze the continuous coinjection of water and gas under steady-state conditions [15,16].

The length over which the mixed zone disappears from the injection well is known as the segregation length ($L_g$). Two equations for $R_g$ (in a cylindrical reservoir) or $L_g$ (in a rectangular reservoir) are derived as:

$$L_g = \frac{Q_t}{k_z(\rho_w - \rho_g)gW\lambda_{rt}^m},\tag{1}$$

$$R_g = \sqrt{\frac{Q_t}{\pi k_z(\rho_w - \rho_g)g\lambda_{rt}^m}},\tag{2}$$

where $\lambda_{rt}^m$ denotes the total relative mobility in the mixed zone, and the parameters $Q_t$, $k_z$, $\rho_w$, $\rho_g$, $g$, and $W$ represent the total volumetric injection rate of gas and water, vertical permeability, densities of water and gas gravitational acceleration, and the thickness of the rectangular reservoir perpendicular to flow, respectively. The extent of volumetric sweep efficiency rises with higher values of $L_g$ or $R_g$, which are influenced by the total injection rate $Q_t$.

Several researchers have performed numerical simulations in order to comprehend the underlying mechanisms involved in this process. Ref. [17] suggested that increasing the injection pressure during steady injection into a reservoir is the only way to regulate

gravity segregation. Furthermore, the ramifications of this model for foams utilized in practical field applications were deliberated upon. Ref. [18] later provided further insight into the subject and demonstrated that Equations (1) and (2) demonstrate strict accuracy when both steady-state conditions and the standard assumptions of fractional-flow theory are employed. The degree of reservoir heterogeneity exerts an influence on the precision of the segregation length as an indicator of sweep efficiency. Ref. [19] conducted a study that showcases the satisfactory accuracy of Equations (1) and (2) when applied to mildly heterogeneous reservoirs characterized by layers or checkerboard patterns. In order to properly account for the heterogeneity, it is necessary to modify the vertical permeability accordingly. Nevertheless, in instances of heightened reservoir heterogeneity, the effectiveness of $L_g$ as a measure of sweep efficiency may be compromised. Additionally, as foam can be used to regulate gravity segregation during coinjection, extending this model to foam flow may be a useful tool for optimizing water–gas coinjection and improving sweep efficiency. This model was expanded to encompass the phenomenon of foam flow, given the condition of uniform injection along the wellbore, notwithstanding the intricate nature of foam behavior [20].

The gravity-segregation model developed by Stone has been extended to dipping reservoirs through numerical simulations [21–23] or an analytical model that makes some assumptions that may not be strictly accurate [24,25]. Nevertheless, limited investigations were conducted concerning the impact of reservoir heterogeneity on the phenomenon of gravity segregation in foam-enhanced oil recovery (EOR) procedures. In this work, we employ an implicit-texture (IT) model, called the 'STARS' model, which assumes local steady-state conditions to simulate foam generation and destruction [26]. The model integrates the influence of foam bubbles implicitly through the inclusion of a mobility-reduction factor, which is dependent on various parameters, including water/oil saturation, capillary number (accounting for shear effects), surfactant concentration, and salinity. The oil phase is not considered here and we assume that surfactant exists within the aqueous phase across the entire porous medium. Appendix A provides a detailed description of the 'STARS' foam model used in this work.

This paper aims to investigate key problems related to gravity segregation in foam-application processes. Firstly, we provide a brief overview of these problems of interest. Then, we verify the accuracy of our simulation framework by comparing the numerical results with analytical solutions for a homogeneous porous medium. We also examine the impact of numerical dispersion on gravity segregation. In addition, we investigate the nature of gravity segregation in layered porous media, including the ultimate segregation length and storage efficiency. Finally, we conclude the paper by discussing and summarizing the key findings.

## 2. Problem Description

Ref. [15] demonstrated that vertical flow barriers, for instance, low-permeability zones, can result in increased recovery compared to a uniform reservoir. However, the impact of barriers or preferential-flow channels on gravity segregation during foam injection remains unknown. This study aims to examine how reservoir heterogeneity affects gravity segregation in foam-application processes in a two-layer reservoir, as depicted in Figure 2, by varying layer thicknesses ($H_1$ or $H_2$), permeabilities ($K_1$ or $K_2$), and foam parameters (*fmmob*, *fmdry*, *epdry*). This model originates from a realistic layered reservoir with a thickness of around 30 m. The permeability ratios are changed, resulting in different thicknesses for each layer, depending on the thickness ratio $R$. The foam parameters for different rock types are determined based on experimental data fitting.

The foam parameters for different rock permeabilities were obtained from a study by [27], wherein the influence of rock permeability on the strength of $CO_2$ foam was examined. The permeability range investigated in their study varies from 32.8 mD to 551.5 mD. The characteristics of the implicit-texture foam model employed in this investigation are comprehensively described in detail in Appendix A. The foam-quality scans were fitted

using a least-squares method [28], as shown in Table 1. To simplify the problem, we made the assumption that the wettability of rocks remains unaffected by permeability, alongside postulating that the Corey exponents and residual saturations in water–gas relative permeability exhibit uniformity across all formations. With these assumptions, the foam parameters for the implicit-texture model were derived.

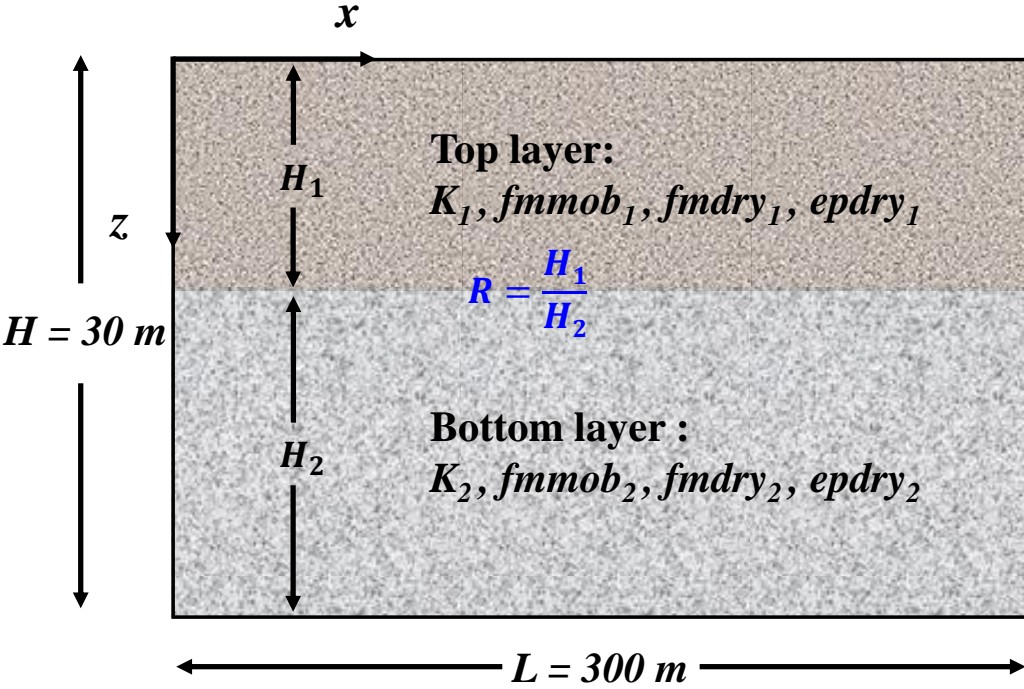

**Figure 2.** Schematic two-layer model used in this work.

Figure 3 illustrates the correlation between the gas apparent viscosity and foam quality, pertaining to rocks characterized by diverse permeabilities. The plot exposes two distinct regimes for all permeabilities. Within the low-quality range, the apparent viscosity of foam experiences a progressive increment and eventually attains a peak value as the foam quality increases. However, within the high-quality regime, the apparent viscosity of the foam exhibits a declining trend with the progressive increase in foam quality. Additionally, the results indicate that foam apparent viscosity rises with increasing rock permeability, signifying that foam is stronger in higher-permeability rocks.

**Table 1.** Relative-permeability and foam parameters used in this work.

| | Permeability, mD | $k_{rw}^0$ | $n_w$ | $k_{rg}^0$ | $n_g$ | $S_{wc}$ | $S_{gr}$ | $fmmob$ | $epdry$ | $fmdry$ |
|---|---|---|---|---|---|---|---|---|---|---|
| $K_1$ | 32.8 | | | | | | | $1.02 \times 10^3$ | $2.50 \times 10^4$ | 0.185 |
| $K_2$ | 56.8 | | | | | | | $1.58 \times 10^3$ | $9.40 \times 10^3$ | 0.171 |
| $K_3$ | 169.8 | 0.2 | 2.0 | 1.0 | 1.8 | 0.1 | 0.05 | $3.14 \times 10^3$ | $6.96 \times 10^3$ | 0.155 |
| $K_4$ | 551.5 | | | | | | | $9.74 \times 10^3$ | $4.76 \times 10^3$ | 0.136 |

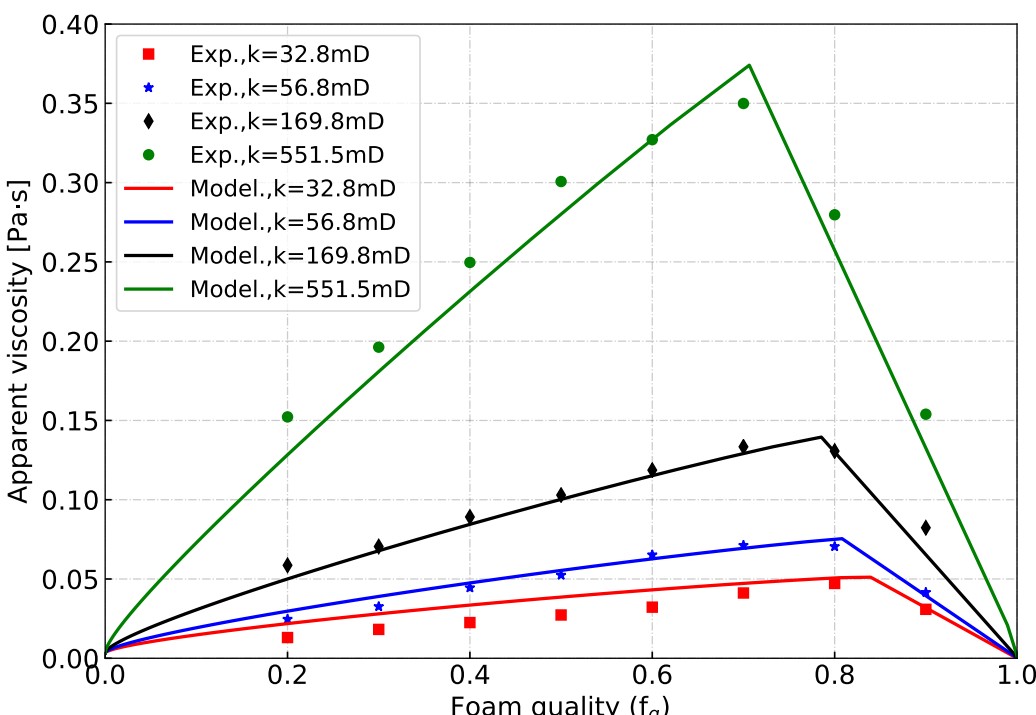

**Figure 3.** Single foam-quality scan of different formations. The symbols in the plots represent the experimental data, while the solid curves depict the results obtained from fitting the data.

## 3. Character of Gravity Segregation in Homogeneous Porous Media

### 3.1. Gas–Water Coinjection and Foam Injection

A two-dimensional horizontal rectangular grid, with each grid block of $1 \times 1 \times 1$ m, is constructed to validate the simulations. Numerical results are compared to an analytical model for a uniform horizontal reservoir with a permeability of 56.8 mD. Both foam injection and gas–water coinjection are tested with varying injected water fractions. No-flow boundaries are made at the top and bottom surfaces of the model, while the left boundary is characterized by injection wells that maintain an injection rate of 1.2 m$^3$/day under reservoir conditions. The segregation length is estimated to be around $10^3$ m for these four sandstones with different permeabilities, assuming a total injection rate of 1.2 m$^3$/day with a water fraction ($f_w$) of 25%, based on Equation (1). However, to better observe the foam behavior within a shorter distance, the injection rate thus is reduced to 0.12 m$^3$/day with $f_w = 25$%, except where noted. The production well is situated along the right boundary, perforated along its length, and has a fixed bottom-hole pressure of 138 bar. To ensure uniformity of $f_w^J$ throughout the reservoir, separate injection wells with a constant injection rate and $f_w$ are used for simulations in each grid block. The study neglects capillary pressure, assuming isotropy of the reservoir (equal horizontal and vertical permeability). Previous studies [15,29] suggest that one pore volume (PV) of gas injection proves adequate in attaining a steady state. However, in this study, two PVs of gas are injected to achieve complete segregation of the injected gas and water. Eight cases are tested with varying injected water fractions of 5.0%, 10.0%, 15.0%, 20.0%, 25.0%, 30.0%, 35.0%, and 40.0%.

The water-saturation profile under steady-state conditions is depicted in Figure 4 for the case where gas and water are coinjected uniformly throughout the entire vertical interval with $f_w = 25$%. The profile consists of a mixed zone exhibiting nearly uniform saturation, a steady-state override region wherein water persists at its residual saturation, and an underride zone characterized by pure water ($S_w = 1$). However, when foam is injected, notable differences can be observed. Firstly, the water saturation within the mixed zone exhibits a significant decrease when foam injection is implemented compared to the

absence of foam, accommodating the injected fractional flow of water. It can be attributed to the reduced gas mobility induced by foam, leading to an elevation in injection pressure despite a constant injection rate. It is noteworthy that even with a tenfold increase in the total injection rate, the injection pressure difference with foam injection (∼70 bar) is approximately 7 times greater than without foam (∼10 bar). The augmented apparent viscosity of foam has the potential to amplify the injection pressure, surpassing the fracture pressure of the formation or limits imposed by surface facilities [30–32]. In addition, the introduction of foam injection at identical rates results in a substantial increase in the segregation length, surpassing a tenfold magnitude extension. According to Equation (1), the effect of gravity segregation is mitigated, because foam injection results in a reduction in the overall mobility of the mixed zone. However, the aforementioned result is accompanied by the consequence of heightened injection pressure.

In Figure 5, a comparison is presented between the analytical and numerical solutions for the segregation length with and without foam, for different injected water fractions. Figure 5a shows the disparity in segregation length when considering the presence or absence of the transition zone. The mixed zone is identified at a saturation of $S_{w,mix}$ based on Stone's model. To mitigate the effects of numerical dispersion, the delineation of the mixed zone is established based on the criterion of water saturation being equivalent to or lower than ($S_{w,mix} + 0.001$). The segregation length is significantly affected by numerical dispersion for the case with the absence of foam, which is discussed below. Figure 5b depicts the considerable influence of foam quality on the segregation length. This impact is achieved through the modification of the aggregate relative mobility within the mixed zone, as described in Equation (1). In the high-quality regime, a decrease in foam quality results in an increase in the segregation length, whereas in the low-quality regime, the opposite trend is observed. The point of transition between the two regimes, characterized by the peak apparent viscosity of the foam, is accompanied by the attainment of the maximum segregation length value, which corresponds to the lowest total relative mobility. The numerical model employed in our study exhibits a commendable level of concurrence with the analytical solutions proposed by [15,16], disregarding the numerical dispersion impact.

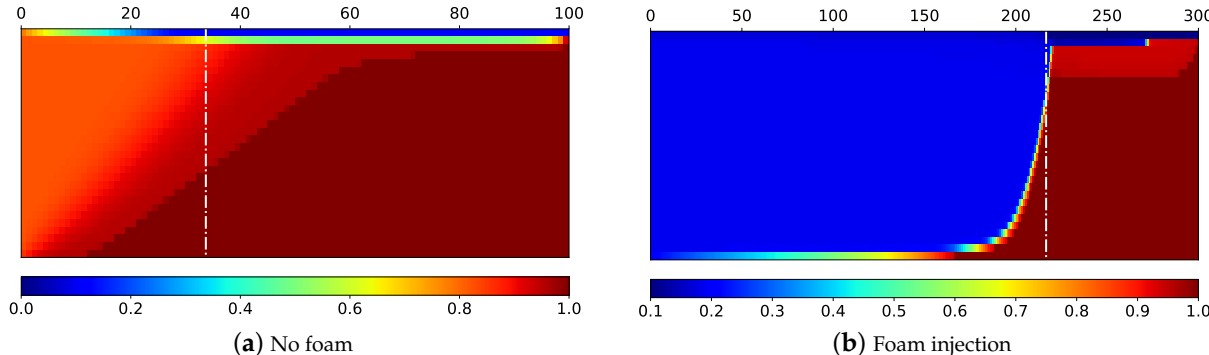

**(a)** No foam  **(b)** Foam injection

**Figure 4.** Distribution of water saturation ($f_w$ = 25%) in a homogeneous porous medium under steady state, with the white dashed line indicating the predicted segregation point by Equation (1). Notably, both cases exhibit transition regions characterized by reduced water saturation compared to the initial state.

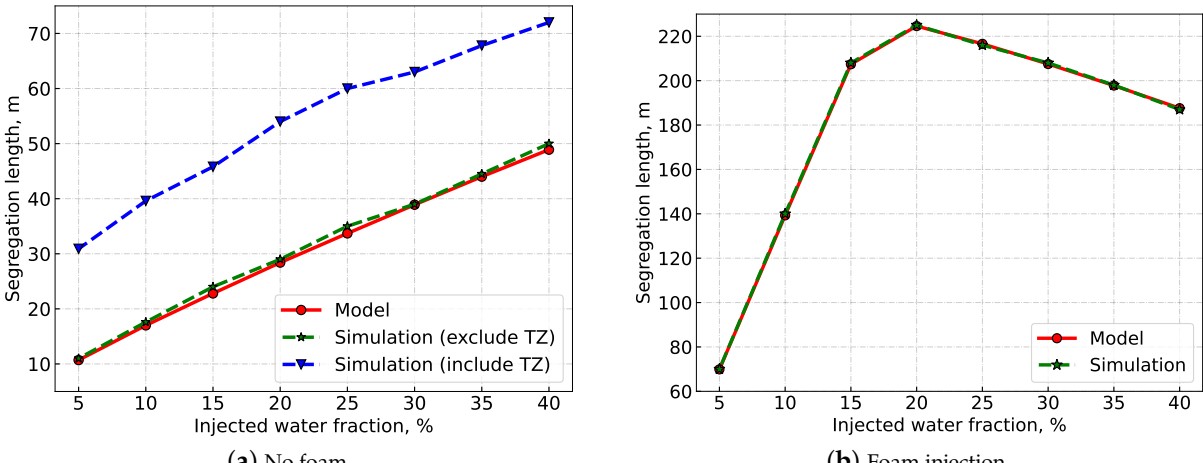

**Figure 5.** An evaluation of the concordance between an analytical model and simulation outcomes. Injection rate is 10 times less for the foam case. (**a**) foam absent; (**b**) foam present. 'TZ' is an abbreviation of 'Transition Zone'.

Ref. [20] demonstrated a significant relationship between injection pressure and the phenomenon of gravity segregation within the cylindrical flow. Building upon their findings, the current investigation broadens the scope of this correlation's implementation to encompass a rectangular coordinate system, as outlined in Appendix B. The correlation provides an upper limit on the injection pressure required for the onset of gravity segregation

$$p(r_w) - p(L_g) = \frac{L_g^2 k_z (\rho_w - \rho_g) g W}{k_h A}, \tag{3}$$

and the lower bound is

$$p(r_w) - p(L_g) = \frac{L_g^2 k_z (\rho_w - \rho_g) g W}{2 k_h A}, \tag{4}$$

where $p(r_w)$ and $p(L_g)$ denote the pressure at the wellbore and at the segregation point, respectively. Equations (3) and (4) do not involve either $Q$ or $\lambda_{rt}^m$, implying that the extent of the segregation point is exclusively governed by the pressure exerted at the injection well, in agreement with the observations made by [20]. Additionally, the pressure difference between the injection point and the segregation point in the upper bound is twice that in the lower bound. These observations are elaborated in Appendix B. Table 2 presents a comparative analysis between the injection pressure values derived from the model and simulations. The mixed zone's shape is depicted in Figure 4b, which closely resembles the case where the mixed-zone height decreases proportionately with the total flow rate, corresponding to the lower-bound assumption described in Equation (4). It is noteworthy that the injection pressure obtained from simulations closely approximates the lower threshold. However, it is important to acknowledge that using this approximation, i.e., based on the mixed zone's shape, may lead to deviations while predicting the injection pressure in this study. Therefore, it is essential to develop a model that accurately characterizes the correlation between injection pressure and the morphology of the mixed zone. Furthermore, it should be emphasized that attaining an equivalent segregation point using stronger foam necessitates an elevated injection pressure.

**Table 2.** An analysis of injection pressure variations across diverse models.

| Case | Model (Lower), Bar | Model (Upper), Bar | Simulation, Bar |
|---|---|---|---|
| K = 32.8 mD | 204.0 | 308.4 | 238.4 |
| K = 56.8 mD | 192.9 | 259.8 | 212.5 |
| K = 169.8 mD | 154.3 | 179.6 | 168.2 |
| K = 551.5 mD | 141.7 | 169.8 | 160.8 |

*3.2. Role of Transition Zone*

In order to examine the underlying reasons for the transition zones depicted in Figure 4, a series of small 2D simulations were carried out to analyze the influence of grid resolution. To achieve higher resolution, we used a domain size of 60 m × 1 m × 20 m without foam injection, and varied the grid block sizes. The grid block sizes in this case are 0.1 m × 1.0 m × 0.1 m, 0.5 m × 1.0 m × 0.5 m, and 1.0 m × 1.0 m × 1.0 m, respectively. When foam was present, we used a domain size of 300 m × 1 m × 30 m. Based on our findings, it can be inferred that the introduction of a surfactant leads to a negligible distinction between the transition area connecting the mixed and underride zones. As such, the investigation solely involves altering the vertical grid block size ($dz$) to analyze the impact of grid size on the transition zone located beneath the override zone when foam is present. The injected $f_g$ of 75% remained constant across all experimental cases. In accordance with the information presented in Table 1, the remaining parameters remain constant.

According to [16], the model offers a technique for calculating the steady-state thickness of the over-/underride zone. This calculation is expressed as follows:

$$\frac{H_w}{H_g} = WAG \frac{\lambda_{gg}}{\lambda_{ww}} = \frac{Q_w}{Q_g} \frac{\lambda_{gg}}{\lambda_{ww}}, \tag{5}$$

where $H_w$ and $H_g$ represent the thickness of the underride zone and override zone, respectively. Additionally, $\lambda_{gg}$ and $\lambda_{ww}$ denote the relative mobility of gas in the override zone and the relative mobility of water in the underride zone, respectively. According to [16], the parameter $\lambda_{gg}$ can be understood as the gas mobility when irreducible water saturation is reached, while $\lambda_{ww}$ represents the mobility of water at a saturation level of 100%. The determination of the water-alternating-gas (WAG) ratio entails the consideration of the volumetric injection rates of water ($Q_w$) and gas ($Q_g$).

Gas saturation profiles at steady-state for different grid sizes in the absence of surfactant are depicted in Figure 6. The segregation length across different cases varies because of the change in grid resolution. The interface between the mixed and underride zones creates a region of transition where the presence of gas is not anticipated in the model under consideration. Based on Equation (5), the height of the override zone is estimated to be approximately 1.32 m. The analytical and numerical results for the segregation point and override zone thickness differ significantly for a grid block size of $dz$ = 1.0 m. These differences decrease with a finer grid resolution, as demonstrated in Figure 6a for a grid block size of 0.1 m. The numerical dispersion effect is responsible for the transition zones when the surfactant is absent. Despite the insignificant disparity in total relative mobility behind the shock front, numerical simulations inadequately capture shocks. Increasing the grid resolution decreases the deviation from analytical results, but it comes at a higher computational cost.

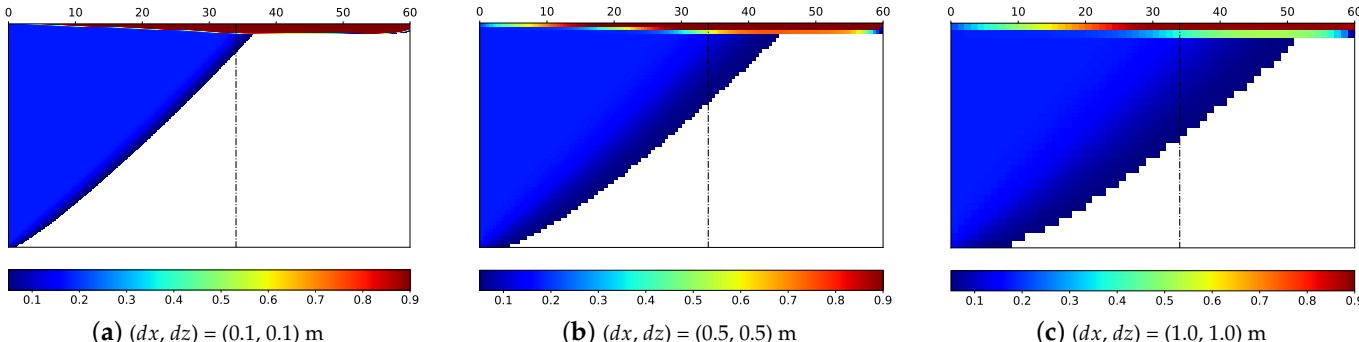

**Figure 6.** Gas saturation profiles without the presence of foam under steady-state conditions, considering different grid sizes. The white region indicates the portion where gas saturation remains below the residual gas saturation. The predicted segregation point based on Equation (1) can be identified as the dashed black line on the profiles.

Figure 7 depicts the gas saturation profile (with surfactant present) at steady state under varying grid sizes. Notably, the introduction of foam into the formation significantly reduces gas mobility, resulting in a negligible transition region between the mixed and underride zones. With foam injection, every grid block traverses a saturation range from the injection condition (shock) to the initial condition. At the intermediate saturation, the total mobility lies between the values at the initial and injected saturations. In the region of override, gas migrates upwards towards the upper layers and undergoes accumulation ahead of foam, thus weakening the foam until it collapses completely. The aforementioned process induces the migration of numerous grid blocks via zones with significantly low mobility, thereby leading to a substantial gas diversion towards the underride zone. Stone's model does not account for this effect, and following the attainment of steady-state conditions, a significantly thick transition region persists between the override and underride zones, which remains impervious to grid refinement. The observed effect can be attributed to the authentic consequence of reduced mobility in the propagating wave located aft of the shock [33]. It is postulated that the surfactant compound exists within the aqueous phase across the porous medium; otherwise, the override zone with foam injection would be similar to that observed in gas–water coinjection.

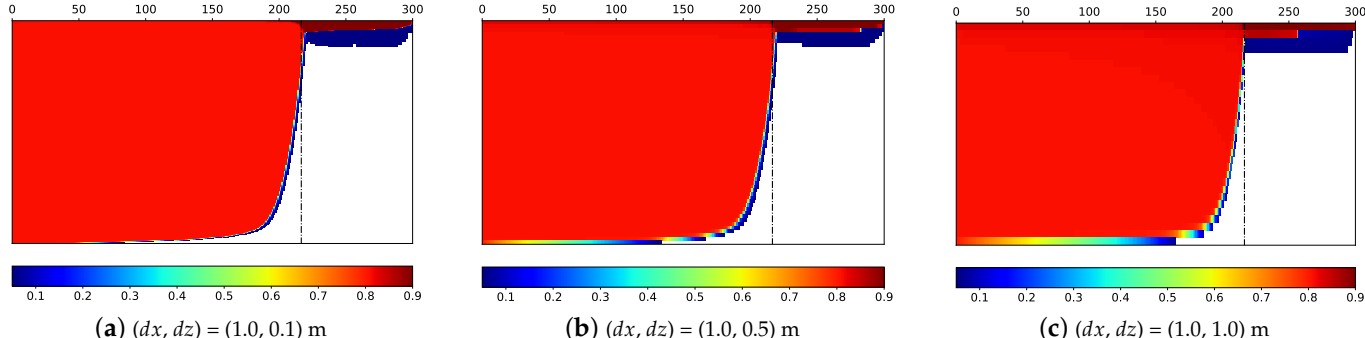

**Figure 7.** Gas saturation profile with foam injection under steady-state conditions with respect to grid-size variations. The white-color region signifies a degree of gas saturation that is lower than the residual gas saturation threshold. The predicted segregation point by Equation (1) is represented by the black dashed line.

## 4. Gravity Segregation in Layered Model

Based on the parameters in Table 1, we construct several two-layer models with different permeability contrasts. It is assumed that there exist no hindrances to flow amidst the various strata. Two injection strategies were formulated: fixed injection rate and fixed injection pressure. In the first case, the total injection rate is fixed at 0.12 m$^3$/day with a

foam quality $f_g$ of 75%, which is bifurcated into two streams to ensure a uniform injection pressure across the entire reservoir (left boundary) predicated on the total mobility in the mixed zone and the thickness ratio between the two strata. The determination of the injection rate for each injection well in each grid block is based on the fraction of the corresponding stream present in that particular layer. In order to facilitate comparative analysis, two homogeneous models with uniform permeability equivalent to either the top layer or bottom layer were used to calculate segregation lengths. The homogeneous models employed in this study exhibit congruity with the two-layer model in terms of the aggregate injection rate, complete thickness, and foam quality. The second injection strategy fixes the injection pressure at 180 bar with foam quality $f_g$ of 75% in each well. This strategy is used to assess the conclusion that injection pressure is the only dominant factor for segregation length in homogeneous porous media.

### 4.1. Effect of Layer Thickness with Fixed Injection Rate

We define a dimensionless parameter $\Omega$ as

$$\Omega = \frac{L_{g,r} - L_{g,H}}{L_{g,L} - L_{g,H}}. \tag{6}$$

Here, the segregation lengths in the two-layer model, higher-permeability homogeneous model, and lower-permeability homogeneous model are denoted as $L_{g,r}$, $L_{g,H}$, and $L_{g,L}$, respectively. The homogeneous model (i.e., single layer) upholds equivalent conditions as the two-layer model, except for the variation in permeability. The parameter represented by the symbol $\Omega$ is utilized to evaluate the influence of the low-permeability layer on the final extent of segregation in a dual-layered system. When the value of $\Omega$ approaches 1.0 in the two-layer model, it indicates that the low-permeability layer dominates the segregation length. Conversely, a value close to 0.0 signifies the dominance of the high-permeability layer in determining the segregation length. The thickness ratio, i.e., the quotient of the top layer thickness ($H_{top}$) to that of the bottom layer thickness ($H_{bottom}$), is also considered to characterize the flow behavior.

The evolution of segregation length under different thickness ratios and various permeability is illustrated in Figure 8. As the thickness ratio increases, it can be observed that the parameter $\Omega$ decreases in the case where the lower-permeability layer is situated at the bottom. In situations where the ratio is small, the low-permeability layer situated at the bottom predominates, whereas $\Omega$ remains constant and approaches 0 as the thickness ratio exceeds a specific value, as illustrated in Figure 8a. In our study, it was observed that for a thickness ratio equal to or exceeding 0.5, the segregation length is predominantly determined by the upper layer with higher permeability. Regardless of the contrast in permeability, the trend observed in $\Omega$ remains almost constant when the layer with higher permeability is situated at the top. By contrast, the value of $\Omega$ exhibits a positive correlation with the thickness ratio and attains 1.0 when the layer of lower permeability is positioned at the top, as depicted in Figure 8b. Nonetheless, there exist ubiquitous findings across diverse scenarios. In cases characterized by a greater permeability contrast (i.e., smaller ratio), there is an increase in the requisite thickness ratio for the lower-permeability layer to assert its dominance, thereby magnifying the impact of the higher-permeability layer situated beneath. Conversely, if the higher-permeability layer is positioned at the top, $\Omega$ exhibits deviating from 1.0 (Figure 8a), which suggests that the presence of the thin layer with high permeability at the top has an impact on the ultimate segregation of the thick layer with low permeability. Similarly, in cases where the layer with lower permeability is situated at the topmost position, the value of $\Omega$ does not equate to zero (Figure 8b). The observed increase in the ultimate segregation length aligns with the inference that a low-permeability stratum at the uppermost region of the reservoir can effectively bolster sweep efficiency. (i.e., resulting in a greater segregation distance) [15,19].

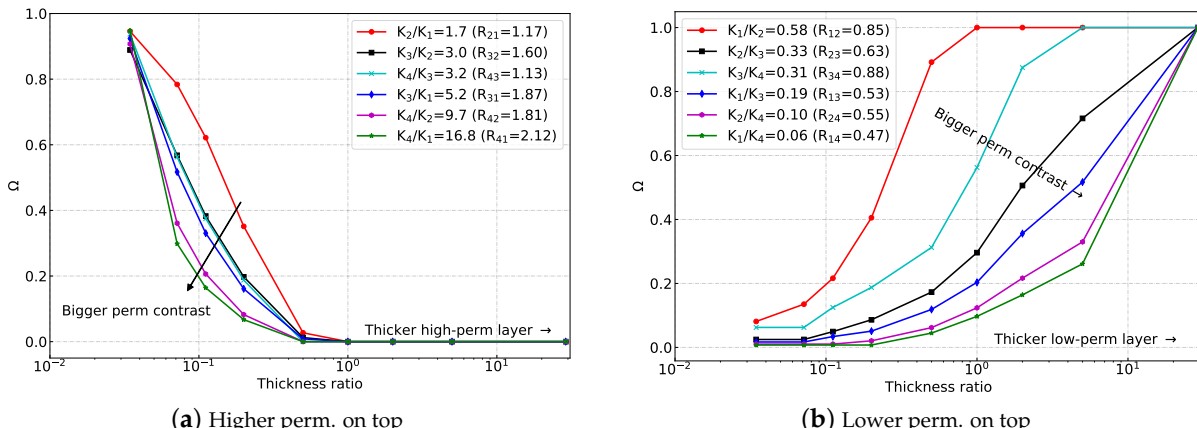

(**a**) Higher perm. on top

(**b**) Lower perm. on top

**Figure 8.** Dependence of the gravity-segregation parameter $\Omega$ on the permeability ratio and thickness ratio where the permeability values are arranged in ascending order as $K_1 < K_2 < K_3 < K_4$. The total mobility ratio in the two layers is denoted by $R_{ij}$.

Figure 9 shows the average injection rate (i.e., injection rate per meter in each layer) in each case. The average injection rate in each configuration is considerably different. With a bigger permeability contrast, the average injection rate into the high- and low-permeability layer differs significantly: for instance, for permeabilities $K_1$ and $K_4$. Such significant differences affect the segregation process in different layers. If the permeability contrast is mild, the average injection rate is similar in different layers. The final segregation length follows similar trends whether the high- or low-permeability layer lies on the top (see Figure 8 red curves). Figure 9 also demonstrates that the cause of $\Omega < 1$ (Figure 8a) and $>0$ (Figure 8b) when a thinner layer is on the top. A thin high-permeability on the top provides a preferential-flow path and slightly reduces the injection rate in the low-permeability bottom layer. However, a thin low-permeability on the top mitigates the process of gas moving upwards and increases the injection rate into the high-permeability layer.

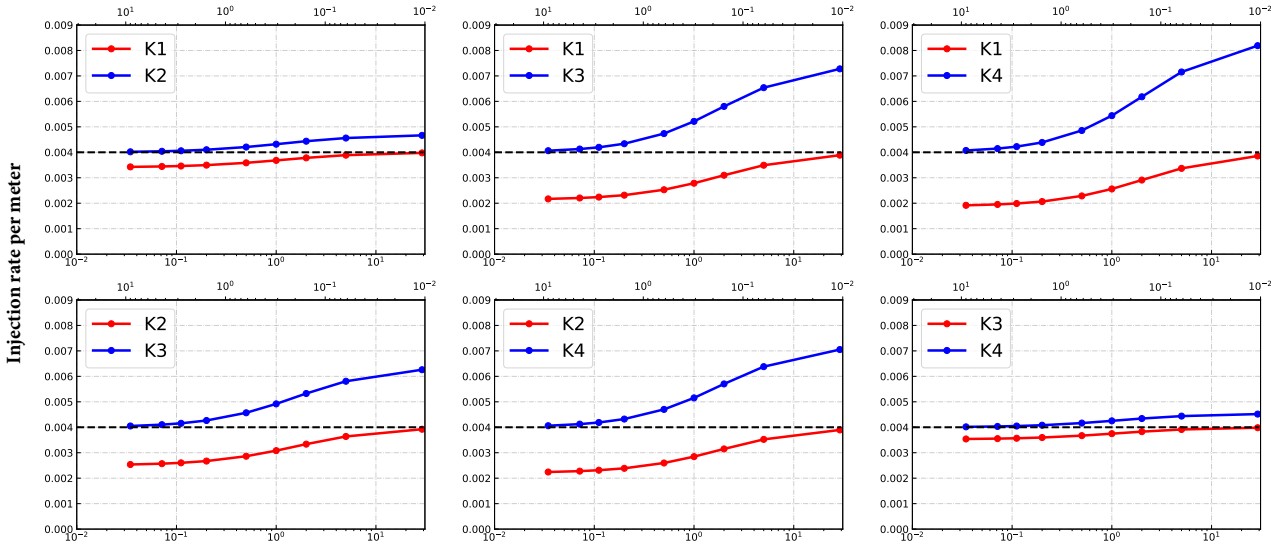

**Thickness ratio (bottom x axis: low-perm layer is on the top; top x axis: high-perm layer is on the top)**

**Figure 9.** Injection rate per meter in each case. The black dashed line is the injection rate in the uniform model, and it is a constant. Two $x$ axes are added to distinguish the thickness ratio with different permeability distributions. In each of the depicted figures, the lower $x$ axis denotes the scenario wherein the low-permeability layer occupies the uppermost position, whereas the upper $x$ axis portrays the situation in which the high-permeability layer is positioned at the top.

### 4.2. Variations of Injection Pressure with Fixed Injection Rate

Ref. [20] pointed out that for homogeneous reservoirs, the gravity segregation length at steady state is dominated by the injection pressure (Appendix B). As mentioned above, the overall injection rate is segregated into two distinct streams based on the injectivity characteristics of each layer; therefore, the injection pressure is uniform along the left boundary, theoretically. Three permeability contrasts (extreme ratios, either large, small value, or moderate value) are chosen to examine the influence of injection pressure on gravity segregation.

Figure 10 shows the injection pressure in different scenarios with various permeability contrasts and thickness ratios. The injection pressure varies significantly among the cases despite a fixed injection rate, owing to differences in injectivity (i.e., total mobility in the two layers). For the cases where the permeability contrast is greater than 1, the injection pressure decreases with increasing thickness of the top layer. Similarly, $\Omega$ also decreases, indicating that the injection pressure dominates the segregation process. If the thickness ratio is fixed, the injection pressure is greater in a case with a moderate permeability contrast, compared to that in an extreme permeability contrast. Permeability distribution is a key factor that affects the ultimate segregation length. The case with permeabilities $K_1$ and $K_4$ shows that the high-permeability top layer completely dominates the segregation with a thickness ratio above 1.0. Conversely, the low-permeability layer starts to influence the segregation though the injection pressure of both cases is close (Figures 8 and 10).

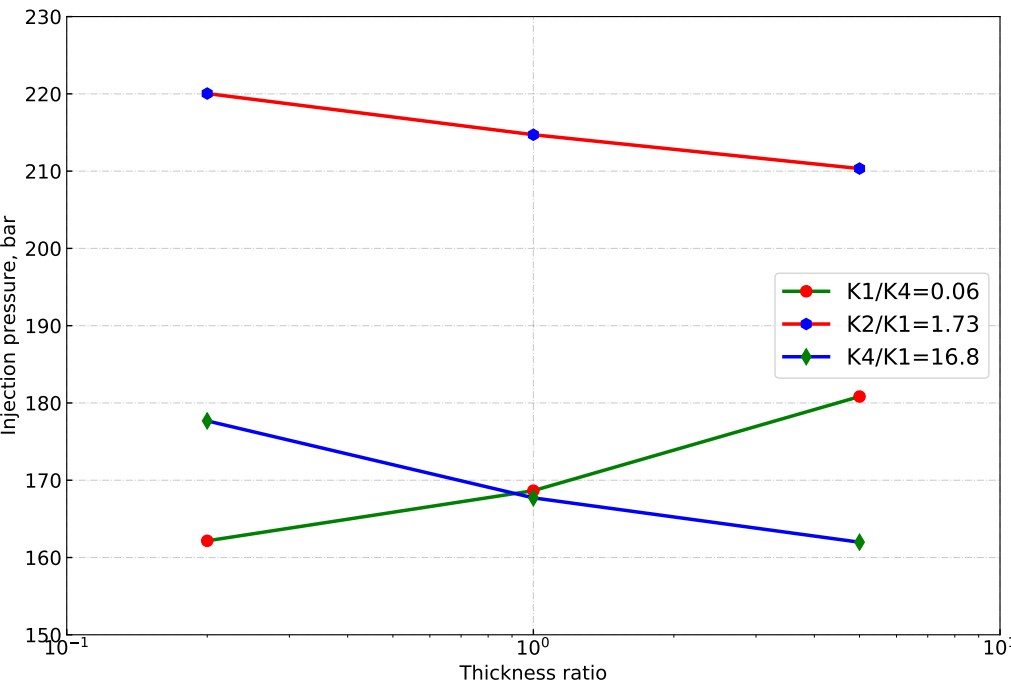

**Figure 10.** Variations of injection pressure in each case with different permeability contrast and thickness ratio. $K_1 / K_4$ represents the low-permeability layer on the top. The thickness ratio is defined as $H_{top} / H_{bottom}$.

To further understand the underlying physics behind the segregation process in layered reservoirs, the water saturation during a transient displacement into two layers of equal thickness is shown in Figures 11 and 12. Gas and water move faster in the high-permeability layer. The progression of a moving front in a high-permeability layer is observed to be advanced in comparison to its progression in a low-permeability layer, if the low-permeability layer is positioned at the top. The gas from the underlying high-permeability layer is transported into the overlying layer characterized by low-permeability, ultimately commingling with the progressing fluids. After gas breakthrough on the upper layer, gas is able to migrate upward with ease while water descends, and a steady state is

gradually established. In this process, the overall moving front is discontinuous. However, a high-permeability layer on top mitigates the velocity difference along the interface between two layers, leading to a smooth-moving front (Figure 12). The thickness of the overlying zone featuring a high-permeability layer positioned atop exhibits a thinner magnitude compared to the configuration wherein a low-permeability layer occupies the topmost position. It indicates that in practice with foam injection following a large surfactant preflush, the displacement efficiency between the override and underride zones is also greater if a low-permeability layer is present at the top except for the extension of segregation length.

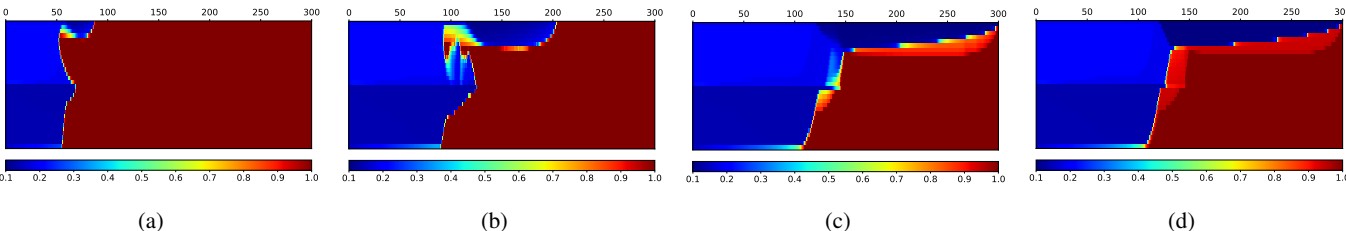

|  (a)  |  (b)  |  (c)  |  (d)  |

**Figure 11.** Temporal evolution of water saturation in transient flow conditions, with a specific focus on the scenario where the low-permeability layer is positioned at the top. The permeability contrasts between the layers are quantified by the ratio of $K_1$ to $K_4$, while both layers possess equal thicknesses. (**a**–**d**) is 0.172 PVI, 0.344 PVI, 0.688 PVI, and 4.0 PVI gas injection, respectively.

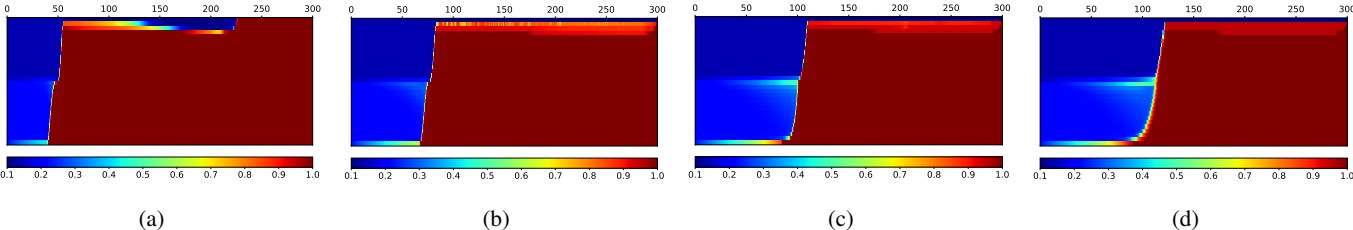

|  (a)  |  (b)  |  (c)  |  (d)  |

**Figure 12.** Temporal evolution of water saturation in transient flow conditions, with a specific focus on the scenario where the high-permeability layer is positioned at the top. The permeability contrasts between the layers are quantified by the ratio of $K_1$ to $K_4$, while both layers possess equal thicknesses. (**a**–**d**) is 0.172 PVI, 0.344 PVI, 0.688 PVI, and 4.0 PVI gas injection, respectively.

### 4.3. Sweep Efficiency with Two Injection Strategies

In this part, we compare the sweep efficiency between two injection strategies in the layered reservoir: fixed injection pressure and fixed injection rate with foam quality of 75% in all cases. The ratio of the volume occupied by $CO_2$ foam to the total pore volume is commonly used to quantify the sweep efficiency. The relative-permeability model utilized in this study is inadequate in characterizing the behavior of trapped gas in the transition zone located below the override zone. The gas saturation should be much higher there because the presence of foam in that zone can increase the trapped gas saturation. Therefore, we assume that 70% of gas is trapped in the transition zone [34].

Figure 13 shows that the sweep efficiency with different injection strategies is obviously distinct. With a thin low-permeability top layer, the sweep efficiency increases if the injection rate is fixed (Figure 13a), consistent with the conclusion in Stone [15]. As the thickness of the low-permeability upper layer increases, the associated sweep efficiency rises, nearing the magnitude of homogeneous porous media with low permeability. A significant contrast in permeability has the potential to result in a substantial increase in sweep efficiency as the necessary thickness ratio for the dominance of the low-permeability layer is amplified. Conversely, an increase in the thickness of the high-permeability top layer results in a decrease in sweep efficiency, indicating that the presence of a high-permeability top layer has an adverse effect on sweep efficiency. However, once the top

high-permeability layer starts to dominate the segregation, the sweep efficiency rebounds and gets close to the value of uniform porous media with high permeability.

If the injection pressure is fixed in the layered model, a thin low-permeability layer on the top to extend the sweep efficiency occurs only when the permeability contrast is not huge (Figure 13b). Given a significant contrast in permeability, the impact of a thin top layer exhibiting low permeability may be disregarded as the quantity of gas present in the low-permeability layer is insufficient to influence the segregation within the underlying layer with high permeability. The sweep efficiency diminishes as the top low-permeability layer thickness increases, owing to the decreased injectivity of the entire layer, while the sweep efficiency increases significantly with a thicker high-permeability top layer.

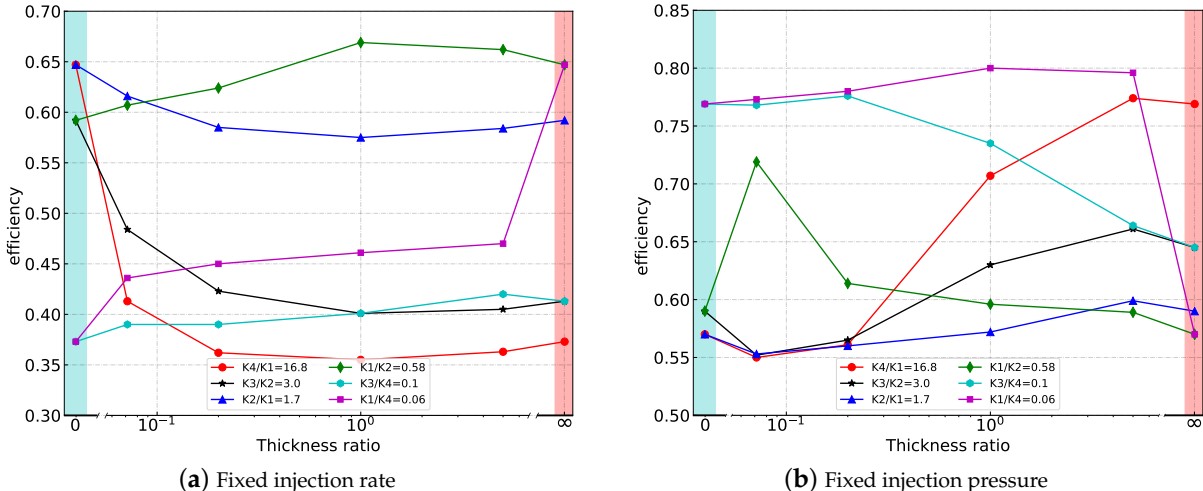

(**a**) Fixed injection rate  (**b**) Fixed injection pressure

**Figure 13.** Sweep efficiency with different injection strategies in layered reservoirs. The light green and light red regions represent the sweep efficiency of homogeneous porous media.

### 4.4. Comparison to Model with Average Properties

There is no rigorous mathematical proof for precise prediction of segregation length with foam injection in heterogeneous reservoirs. Following [19] work, we approximately calculate the segregation length in two-layer models by adjusting $K_v$ to represent the effective vertical permeability. The harmonic average permeability for the layered reservoirs is used:

$$\overline{K_{v,ave}} = \frac{h_t}{\sum_{i=1}^{n} \frac{\Delta h_i}{K_{v,i}}}, \tag{7}$$

where $n$ is the number of layers in the reservoir, $h_t$ is the total thickness, and $h_i$ and $K_i$ are the thickness and vertical permeability of layer $i$, respectively.

After obtaining the $\overline{K_v}$, we directly linearly interpolate the foam parameters based on Table 1. Then we can approximate the segregation length in these layered reservoirs by running simulations. The results are shown in Table 3. With a larger thickness ratio, i.e., the average permeability is close to that of the top layer, the model possessing average properties demonstrates the capability to predict the distance of complete segregation with precision in scenarios where the layer with higher permeability is positioned at the top. However, if the thickness of the two layers is not significantly different, the approximation overestimates or underestimates $L_g$ depending on which permeability is on the top. When heterogeneity is more severe, the average properties cannot be used to predict the steady-state segregation length accurately.

**Table 3.** Comparison of gravity-segregation length in different formations.

| Thickness Ratio, m/m | $K_1$ & $K_2$ Combination | | | | | $K_1$ & $K_3$ Combination | | | | | $K_1$ & $K_4$ Combination | | | | |
|---|---|---|---|---|---|---|---|---|---|---|---|---|---|---|---|
| | Ave. $K$, mD | Ave. $\lambda_{rt}$, 1/Pa·s | $L_{seg}$, m | $L_{seg,D}$, m | $L_{seg,I}$, m | Ave. $K$, mD | Ave. $\lambda_{rt}$, 1/Pa·s | $L_{seg}$, m | $L_{seg,D}$, m | $L_{seg,I}$, m | Ave. $K$, mD | Ave. $\lambda_{rt}$, 1/Pa·s | $L_{seg}$, m | $L_{seg,D}$, m | $L_{seg,I}$, m |
| 1/29 | 33.6 | 40.01 | 158 | 158 | 135 | 37.4 | 37.12 | 153 | 158 | 83 | 50.1 | 30.50 | 139 | 158 | 73 |
| 5/25 | 36.8 | 37.73 | 153 | 145 | 155 | 55.6 | 28.09 | 136 | 100 | 93 | 119.3 | 18.94 | 94 | 80 | 80 |
| 10/20 | 40.8 | 35.18 | 148 | 135 | 158 | 78.5 | 23.74 | 114 | 87 | 111 | 205.7 | 12.75 | 81 | 74 | 93 |
| 15/15 | 44.8 | 32.93 | 144 | 135 | 158 | 101.3 | 20.76 | 101 | 83 | 136 | 292.2 | 9.44 | 77 | 73 | 109 |
| 20/10 | 48.8 | 31.09 | 140 | 135 | 158 | 124.1 | 18.40 | 93 | 83 | 158 | 378.6 | 7.48 | 75 | 73 | 129 |
| 25/5 | 52.8 | 29.37 | 137 | 135 | 158 | 147.0 | 16.61 | 87 | 83 | 158 | 465.1 | 6.17 | 74 | 73 | 154 |
| 29/1 | 56.0 | 28.10 | 135 | 135 | 158 | 165.2 | 15.49 | 83 | 83 | 158 | 534.2 | 5.45 | 73 | 73 | 158 |

$L_{seg}$ is the segregation length of homogeneous reservoir with uniform properties, $L_{seg,D}$ and $L_{seg,I}$ are the corresponding two-layer model where high- and low-permeability layer is on the top, respectively. $K_1 < K_2 < K_3 < K_4$.

## 5. Summary and Discussion

The assessment of foam-based enhanced oil recovery (EOR) relies significantly on the gravity-segregation length ($L_g$) as predicted by Stone's model, underscoring its importance as a key parameter. It highly depends on permeability distribution. In turn, the heterogeneity affects foam strength. In horizontal reservoirs with uniform and isotropic permeability, foam can effectively mitigate the gravity-segregation process and improve sweep efficiency. In practice, the distance between wells is limited, and the injection rate is much higher. We can infer that the foam strength we used in the field test can be weaker than that obtained from the laboratory, then the injectivity is improved by weakening foam strength.

For layered reservoirs, Stone's model predicts the distance to complete segregation accurately only in cases where the average permeability is close to that of an individual layer. This suggests that the harmonic average permeability for $K_v$ may not adequately represent the effect of heterogeneity on effective vertical permeability. For more severe heterogeneous reservoirs, the analytical model may not be a good tool to predict the segregation length due to lacking criteria to evaluate the vertical permeability.

The impact of heterogeneity on rock-wetting behavior is not considered in our study. This assumption is not rigorously correct because the connate water saturation is usually different in different rocks. In the context of heterogeneous reservoirs, wherein the connate water remains constant, the foam-model parameter, $fmdry$, may exhibit inaccuracies, despite successful experimental data fitting in the low-quality regime.

In addition, the surfactant may not fully fill the reservoir in practice, so the transition zone between the override and underride zones should be close to that in the case without surfactant. This assumption can affect the final sweep efficiency. In the process of foam injection, the extent to which the gas and surfactant propagate can differ based on various factors such as injected quality and adsorption. Additionally, geological heterogeneity is expected to be more intricate than the simplified model utilized in this study, thereby potentially influencing the gravity-segregation process. While these factors have not been accounted for in this research, they should be considered in future investigations.

## 6. Conclusions

This study examines the impact of foam on gravity segregation in reservoirs with varying heterogeneity. Based on our findings, we can conclude that:

- In the horizontal and homogeneous reservoirs, both with and without foam, the numerical results and analytical solutions exhibit a mutually favorable agreement. The length of segregation is dictated by the vertical permeability and the apparent viscosity of foam, which is contingent upon the total mobility at a constant rate of injection.

- The heterogeneity of reservoirs plays a critical role in the regulation of gravity segregation, particularly within two-layer models. The thickness of the uppermost layer exerts a noteworthy influence on the extent of segregation in the underlying layer. A top layer with reduced permeability and thickness results in an elongated segregation

length in the underlying layer. Conversely, the introduction of a top layer with greater permeability and reduced thickness serves to diminish the extent of segregation in the underlying layer. However, for a thicker top layer, the influence of the bottom layer on the segregation length is comparatively less significant.
- Injection pressure is one key factor that affects the ultimate segregation length in a layered model, further affecting sweep efficiency. With different injection strategies, the ultimate sweep efficiency shows a distinct trend.
- A single-layer approximation cannot predict the segregation length in the layered reservoir with a severe permeability contrast and a comparable thickness ratio.

**Author Contributions:** M.W.: methodology, simulation and writing; R.S.: supervision and editing; Y.H.: writing and simulation; D.C.: simulation; Y.L.: writing and editing; H.X.: Conceptualization and editing; W.G.: writing and editing; L.C.: supervision and writing. All authors have read and agreed to the published version of the manuscript.

**Funding:** This research received no external funding.

**Data Availability Statement:** MDPI Research Data policies

**Acknowledgments:** We acknowledge the technical support from Huiqing Liu at the China University of Petroleum (Beijing).

**Conflicts of Interest:** The authors declare no conflict of interest.

## Nomenclature

| | |
|---|---|
| $L_g$ | segregation length in a rectangular reservoir, m |
| $R_g$ | segregation length in a cylindrical reservoir, m |
| $\lambda_{rt}^m$ | total relative mobility in the mixed zone, - |
| $\lambda_{gg}$ | relative mobility of gas in the override zone, - |
| $\lambda_{ww}$ | relative mobility of water in the underride zone, - |
| $Q_t$ | total volumetric injection rate of gas and water, m$^3$/day |
| $Q_w$ | water injection rate, m$^3$/day |
| $Q_g$ | gas injection rate, m$^3$/day |
| $k_z$ | vertical permeability, mD |
| $\rho_w$ | water density, kg/m$^3$ |
| $\rho_g$ | gas density, kg/m$^3$ |
| $g$ | gravitational acceleration, m/s$^2$ |
| $W$ | thickness of the rectangular reservoir perpendicular to flow, m |
| $K$ | absolute permeability, mD |
| $k_{rw}$ | water relative permeability, - |
| $k_{rg}$ | gas relative permeability, - |
| $n_w$ | exponent for water relative permeability, - |
| $n_g$ | exponent for gas relative permeability, - |
| $S_{wc}$ | connate water saturation, - |
| $S_{gr}$ | residual gas saturation, - |
| $epdry$ | parameter controlling the abruptness of foam collapse, - |
| $fmdry$ | limiting water saturation, - |
| $fmmob$ | maximum-attainable gas-mobility reduction, - |
| $fmsurf$ | critical component mole fraction value, - |
| $epsurf$ | exponent for composition contribution, - |
| $W_s$ | component mole fraction, - |
| $p(inj)$ | injection pressure, bar |
| $p(L_g)$ | pressure at the segregation point, bar |
| $A$ | cross-section area, m$^2$ |
| $H_w$ | thickness of the underride zone, m |
| $H_g$ | thickness of the override zone, m |
| $L_{g,r}$ | segregation length in the two-layer model, m |

| $L_{g,H}$ | segregation length in the higher-permeability homogeneous model, m |
| $L_{g,L}$ | segregation length in the lower-permeability homogeneous model, m |
| $h_t$ | total thickness, m |
| $h_i$ | thickness of layer $i$, m |
| $K_i$ | vertical permeability of layer $i$, mD |

## Appendix A. Foam Model

The present study employs the commonly used CMG-STARS model (implicit-texture foam modeling) to examine the impact of water saturation ($S_w$) and surfactant concentration ($W_s$) on foam strength [26,35]. The following factors are considered in the calculations and simulations for the sake of simplicity:

$$k_{rg} = \frac{k_{rg}^0(S_w)}{1 + fmmob F_1 F_2},$$ (A1)

$$F_1 = \begin{cases} \left(\frac{W_s}{fmsurf}\right)^{epsurf} & Ws \leq fmsurf \\ 1 & W_s > fmsurf, \end{cases}$$ (A2)

$$F_2 = 0.5 + \frac{arctan[epdry(S_w - fmdry))]}{\pi}.$$ (A3)

Here, $fmmob$ maximum mobility-reduction factor, $fmdry$ critical water saturation for foam generation and coalescence, $epdry$ exponent for water saturation contribution, $fmsurf$ critical component mole fraction value, and $epsurf$ exponent for composition contribution. $k_{rg}^0(S_w)$ is the function to express the gas-phase relative permeability without foam. As shown in Equation (A1), the gas-mobility experiences a reduction owing to a decline in the gas's relative permeability induced by the presence of foam. It is postulated that foam forms when there is a sufficient quantity of water, gas, and surfactant. In this study, it is postulated that surfactant exhibits homogeneous distribution within the water phase across the porous medium, in order to simplify the analysis.

The physical properties assumed in Equation (1) or Equation (2) are $\mu_w$ = 0.65 mPa·s and $\mu_g$ = 0.05 mPa·s for the dynamic viscosities of water and gas respectively, as well as $\rho_w$ = 1000 kg/m$^3$ and $\rho_g$ = 166 kg/m$^3$ for their respective densities, all of which have been determined by the Redlich-Kwong equation of state (EOS) [36].

## Appendix B. Correlation between Injection Pressure and Segregation Length

Ref. [20] provided empirical evidence supporting a relationship between the injection pressure and the manifestation of gravity segregation in the context of cylindrical flow. Their findings suggest that the sole means of effectively controlling gravity segregation during the continuous injection process into a specific reservoir is through the elevation of pressure at the injection well. In this research, we propose an extension of the previously mentioned association within the context of a Cartesian coordinate system. Specifically, we put forward the conjecture that the bottom-hole pressure of the injection well corresponds to the pressure exhibited by the interconnected reservoir block.

Figure A1 demonstrates two distinct shapes of the mixed, override, and underride zones. In Figure A1a, The proportionality between the quantity of water and gas within the mixed zone is contingent upon the ratio between the position denoted by $x$ and the segregation point represented by $L_g$. Hence, one can deduce that the vertical extent of the mixed zone is dependent on the total rate of fluid flow within the mixed zone. By way of contrast, it can be observed from Figure A1b that the entirety of the gas and water phases persist within the mixed zone until such time that they undergo segregation, leading to a progressive reduction in the mixed zone's height with greater distance from the point of injection, culminating shortly prior to the occurrence of the segregation point. The aforementioned instances depict two distinct thresholds of pressure injection within the water and gas coinjection procedure.

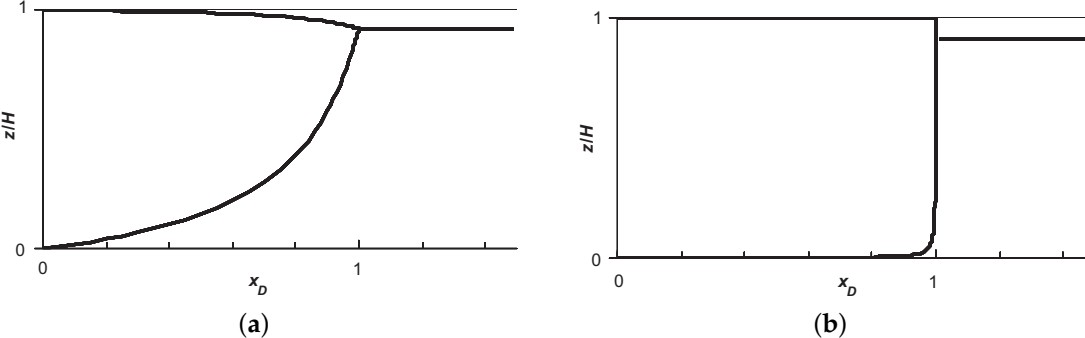

**Figure A1.** Two asymptotic cases of gravity segregation as presented by [20]. (**a**) The mixed-zone height exhibits a near-proportional reduction as the distance from the injection well progressively increases. (**b**) The mixed-zone height demonstrates consistent maintenance as the distance from the injection well progressively increases until it nears the segregation point.

In the first scenario, the overall horizontal superficial velocity, denoted as $U_t$, within the mixed zone of the linear flow, is expressed as a linear equation of the dimensionless ratio $x/L_g$. This relationship suggests that the overall flow rate within the mixed zone diminishes proportionally with it:

$$U_t = \frac{Q}{A}\left(1 - \frac{x}{L_g}\right) = -K_h \lambda_{rt}^m \frac{dP}{dL}, \tag{A4}$$

where $K_h$ is the horizontal permeability and $x$ is position. Through the process of integration, extending from the well to the stage of segregation, it is possible to formulate the pressure required for injection:

$$p(inj) - p(L_g) = \frac{QL_g}{2AK_h\lambda_{rt}^m}. \tag{A5}$$

Here, $p(inj)$ and $p(L_g)$ denote the injection pressure and the pressure at the segregation point, respectively. Subsequently, by reordering the formula presented in Equation (1), an equation for the variable $Q$ can be obtained, which can then be substituted into Equation (A5) to obtain the expression

$$p(inj) - p(L_g) = \frac{L_g^2 K_z (\rho_w - \rho_g) g W}{2AK_h}. \tag{A6}$$

In the second scenario, wherein the mixed zone, the total flow rate is uniform, Equations (A4)–(A6) become

$$U_t = \frac{Q}{A} = -K_h \lambda_{rt}^m \frac{dP}{dL}, \tag{A7}$$

$$p(inj) - p(L_g) = \frac{QL_g}{AK_h\lambda_{rt}^m}, \tag{A8}$$

and

$$p(inj) - p(L_g) = \frac{L_g^2 K_z (\rho_w - \rho_g) g W}{AK_h}. \tag{A9}$$

It should be noted that neither the variable $Q$ nor the parameter $\lambda_{rt}^m$ is incorporated into Equation (A6) or Equation (A9).

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
