# Peer review of "Mitigation of Gravity Segregation by Foam to Enhance Sweep Efficiency"

_applsci, doi:10.3390/app13158622_

Round 1

Reviewer 1 Report

Mitigation of Gravity Segregation by Foam to Enhance Sweep

Efficiency

Thanks for providing me the opportunity to review this interesting paper. My comments are provided below;

·       Please add some key results and numbers to the abstract.

·       Please mention how this model can be beneficial for real-field applications.

·       Please provide the units for all parameters presented in the equations

·       Please add a table to summarize all available models/studies reporting their applicability, assumptions, and limitations.

·       Fig 2, please justify the used numbers, for example why H is 30 m not more or less?

·       Kindly comment on the use of three layers instead of two, what would be the differences?

·       How did you obtain the relative permeability values, please provide a figure to show the Kro and Krw profiles vs. Saturation.

·       Fig 4., what is the color bar indicate? Foam quality?

·       Fig 5, what is the difference between model and simulation? Do you use one of them for the validation?

·       Table 2, how did you select the permeability values? Please just the used numbers.

·       I would suggest improving the label for Figure 7, what are the red and white colors indicate?

·       Please improve the quality of Figure 8.

·       Can you please comment on using this model for radial flow around the wellbore?

·       Please add a table to summarize the obtained results and the applied constraints.

·       Kindly report the work limitations and recommendations for future work.

Reviewer 2 Report

Too long for journal paper, therefore needs for consolidation of the content.  All symbols used should be listed in the nomenclature. Some figures title are too long. Figure title should be short but concise. Some information in the title should be in the related paragraph. There are also some typo error such as in line 329 and 412.

Nil

Reviewer 3 Report

Dear Authors

Congratulation for a very good article.

I only suggest two issues:

1- citing some new related published studies:

https://doi.org/10.1016/j.fuel.2023.128810

https://doi.org/10.1016/j.geoen.2023.211976

https://doi.org/10.1016/j.fuel.2023.129150

https://doi.org/10.1016/j.energy.2023.127860

2- Discussing the machine learning related published approaches.
